# Microvascular Thrombosis and Liver Fibrosis Progression: Mechanisms and Clinical Applications

**DOI:** 10.3390/cells12131712

**Published:** 2023-06-24

**Authors:** Carlo Airola, Maria Pallozzi, Lucia Cerrito, Francesco Santopaolo, Leonardo Stella, Antonio Gasbarrini, Francesca Romana Ponziani

**Affiliations:** 1Hepatology Unit, CEMAD Centro Malattie dell'Apparato Digerente, Medicina Interna e Gastroenterologia, Fondazione Policlinico Universitario Gemelli IRCCS, 00168 Rome, Italy; airollac@gmail.com (C.A.); mariapallozziucsc@gmail.com (M.P.); lucia.cerrito@guest.policlinicogemelli.it (L.C.); santopaolofrancesco@gmail.com (F.S.); leonardo.stella.1991@gmail.com (L.S.); antonio.gasbarrini@unicatt.it (A.G.); 2Dipartimento di Medicina e Chirurgia Traslazionale, Università Cattolica del Sacro Cuore, 00168 Rome, Italy

**Keywords:** ADAMTS-13, coagulation, fibrosis, hepatic stellate cells, liver cirrhosis, microthrombosis, parenchymal extinction, platelets, von Willebrand factor

## Abstract

Fibrosis is an unavoidable consequence of chronic inflammation. Extracellular matrix deposition by fibroblasts, stimulated by multiple pathways, is the first step in the onset of chronic liver disease, and its propagation promotes liver dysfunction. At the same time, chronic liver disease is characterized by alterations in primary and secondary hemostasis but unlike previously thought, these changes are not associated with an increased risk of bleeding complications. In recent years, the role of coagulation imbalance has been postulated as one of the main mechanisms promoting hepatic fibrogenesis. In this review, we aim to investigate the function of microvascular thrombosis in the progression of liver disease and highlight the molecular and cellular networks linking hemostasis to fibrosis in this context. We analyze the predictive and prognostic role of coagulation products as biomarkers of liver decompensation (ascites, variceal hemorrhage, and hepatic encephalopathy) and liver-related mortality. Finally, we evaluate the current evidence on the application of antiplatelet and anticoagulant therapies for prophylaxis of hepatic decompensation or prevention of the progression of liver fibrosis.

## 1. Introduction

Fibrosis is a frequent consequence of organ injury. The formation of an extracellular matrix (ECM) depends on a complex cascade of cellular and molecular pathways, the chronic activation of which results in a sustained fibrogenic process that leads to structural changes and, ultimately, to dysfunction of the affected organ. Thus, fibrosis is a major contributor to organ failure in human pathophysiology [1]. Fibrotic changes are linked to a variety of diseases, suggesting common pathogenetic mechanisms. This “wound response” is controlled by complex cell-specific processes in which distinct molecular pathways are involved [1]. Hemostasis is becoming increasingly important among the many biochemical mechanisms and cellular interactions involved in fibrogenesis.

The development of chronic disease following organ damage has been linked to an imbalance between pro- and anti-coagulant factors. For example, microvascular endothelial cells of the renal parenchyma contribute to the development of a prothrombotic environment in the presence of stressor triggers by increasing the synthesis of prohemostatic factors and reducing the production of protective proteins. Activation of the endothelial surface following the onset of thrombosis and activation of the coagulation cascade induces transformation to a proinflammatory fibrogenic cellular phenotype, which worsens renal damage and causes fibrosis [2]. Additionally, in acute lung injury and fibrotic lung disease, uncontrolled coagulation has been shown to contribute to the dysregulation of inflammatory and fibroproliferative responses [3]. According to pathophysiological evidence, coagulation plays an essential role in many disorders, and fibrosis is often an adverse outcome. Chronic liver disease is the paradigm of conditions in which fibrosis resulting from an acute or chronic insult leads to organ dysfunction. In this review, we aim to investigate the function of microvascular thrombosis in liver disease and to highlight the molecular and cellular networks that link hemostasis to fibrosis in this setting.

## 2. From Microvascular Thrombosis to Hepatic Fibrosis

Studies on the consequences of acute murine hepatitis virus infection have provided the first evidence that coagulation is an important factor in the pathogenesis of liver disease. Indeed, numerous sinusoidal microthrombi directly related to hepatic parenchymal necrosis were described in these animal models, and microvascular thrombosis was associated with more severe hepatitis [4,5,6].

In 1995, examining histological specimens from the livers of patients with chronic heart failure, Wanless et al. found a substantial correlation between sinusoidal fibrosis and the occurrence of local thrombotic events, indicating that liver fibrosis may be the result of microvascular thrombosis. More specifically, it has been proposed that disruption of blood flow by sinusoidal microthrombi and sinusoidal fibrosis cause reactive hyperemia and congestion, which activate fibroblasts and increase collagen deposition, worsening blood flow, inducing extension of thrombosis, neoangiogenesis, and parenchymal necrosis. The end result is the loss of hepatocytes and the formation of fibrous septa that completely alter the architecture of the liver [7]. The replacement of liver parenchyma with fibrotic tissue as a result of microvascular disruption has been called parenchymal extinction [8]. Inflammation is considered to be one of the main mechanisms inducing hepatic fibrogenesis in response to parenchymal damage [9]; however, a preclinical model has shown that mild inflammation at both histological and serological levels is associated with the development of liver fibrosis due to chronic venous congestion [10]. According to this study, sinusoidal thrombosis appears to be crucially involved in the direct potentiation of fibrogenesis in congestive liver disease.

On the other hand, hemostasis has also been firmly linked to the inflammatory response, and recently the concepts of immune-coagulation and thrombo-inflammation have been proposed [11,12]. Several conditions, including ischemia-reperfusion syndrome and infections, can lead to the formation of microvascular thrombosis, which triggers an inflammatory response [13]. As part of inflammatory processes, numerous cell types, including immune cells and fibroblasts, participate in the coagulation process, enhancing inflammation and increasing ECM deposition in different organs [14]. In fact, hemostatic activation could be both the cause and the consequence of the inflammatory process.

However, there are few examples of clinical models of hepatic microvascular thrombosis in the literature, probably because microvascular thrombosis is rapidly replaced by fibrous tissue and is rarely detected on biopsy in patients with chronic liver disorders [8]. Cases of systemic microvascular thrombotic disease in humans suddenly increased following the severe acute respiratory syndrome coronavirus 2 (SARS-CoV-2) pandemic, prompting multiple studies, including some on liver pathology. SARS-CoV-2 causes COVID-19, which is a systemic infectious disease in which endothelial cell dysfunction and microvascular thrombosis are likely to play a major role in the development of multi-organ complications [15]. Although the airways and lungs are the main organs affected by SARS-CoV-2, liver injury is a frequent condition in patients with the most severe clinical forms [16,17]. Sinusoidal microthrombi were frequent in liver biopsy specimens taken from COVID-19 patients and were related to more severe liver damage [18,19]. Mild portal and lobular inflammation, confluent parenchymal necrosis, and fibrosis are other histological findings [18]. Because there is no evidence of direct cytopathic virus damage, the latter appears to be associated with the marked inflammatory activation produced by the infection [20]; in fact, molecular analyses revealed that during severe COVID-19, genes frequently linked to hepatic stellate cells (HSCs) activation and liver fibrosis, such as interleukin 6 (IL6), interleukin 1 (IL1), tumor necrosis factor α (TNF-α), interleukin 10 (IL10), and interferon α (IFN-α), as well as vascular endothelial growth factor (VEGF) and monocyte chemoattractant protein 1 (MCP-1), are overexpressed [21]. In addition, a higher noninvasive fibrosis score appears to be correlated with a higher risk of developing severe COVID-19 [22].

The fascinating link between sinusoidal thrombosis and the activation of molecular pathways of liver fibrosis in humans is supported by the beneficial effects of anticoagulants on the development of fibrosis [23,24]. In line with the direction set out by Wanless et al. in 1995, developments in molecular medicine and the identification of new intra- and intercellular networks have strengthened this connection.

## 3. Hepatic Stellate Cells and Protease Activated Receptor

In response to repeated injury, HSCs can differentiate into myofibroblasts, which proliferate and produce ECM [25]. In addition, HSCs stimulate other cell types that participate in the inflammatory response and fibrogenesis through the production of growth factors and chemokines, playing a critical role in driving the initiation and progression of the fibrogenic process [26]. As expected, microvascular sinusoidal thrombosis, which leads to ischemic injury and inflammatory response, causes overexpression of VEGF, platelet-derived growth factor (PDGF) and transforming growth factor beta (TGFβ) by hepatocytes and HSCs, as well as increased synthesis of type I and type IV collagen by activated HSCs [27]. However, a specific molecular mechanism has been proposed as responsible for the apparent association between microvascular thrombosis and fibrogenesis. In 1998, Marra et al. hypothesized that the protease-activated receptor (PAR) signaling pathway was a major molecular pathway involved in HSCs activation and hepatic scarring [28]. PARs are G-protein-coupled receptors with proteolytic activity that stimulate cellular responses by interacting with coagulation factor Xa (FXa) and neutrophil elastase (PAR 1, 2), thrombin (PAR 1, 3, 4), coagulation factor VIIa (PAR 1), and tryptase (PAR 2, 4), which cleave the N-terminal of the receptor [28]. PARs are expressed by different types of cells involved in the fine regulation of vascular homeostasis, and their signaling pathways are complex because they can be linked to G proteins with different functions. Consequently, they interact with a myriad of signaling transducers (e.g., extracellular signal-regulated kinase [ERK] 1/2, Rho/Rho-kinase, c-Jun N-terminal kinase, inositol 1,4,5-trisphosphate [IP3], phosphoinositide 3-kinases [PI3K], and Janus kinase/signal transducers and activators of transcription [JAK-STAT]), resulting in pleiotropic effects [29,30,31]. PARs have been linked to the progression of fibrosis in several organs, and because they have a high affinity for factors in the coagulation cascade, they have been proposed as the main link between hemostasis and pulmonary fibrosis [32,33] or renal fibrosis [2]. In addition, PAR 1 induces cardiac fibroblast activation in response to thrombin or FXa and, by modulating the ERK1/2 pathway, leads to cardiac remodeling and fibrosis [29,34].

PARs are found in liver cells and are abundantly overexpressed in chronic liver disease, as in the case of PAR 1 in the myofibroblast group of cirrhotic individuals [35]. A preclinical rat liver stellate cell model revealed a progressive increase in PAR 1 and PAR 2 expression during transformation to a myofibroblastic phenotype [36]. Increasing concentrations of thrombin transform HSCs into myofibroblasts increases the production of α-smooth muscle actin (α-SMA), pro-collagen, TGFβ-1, matrix metalloproteinase 2 (MMP-2), and other cellular signals essential for wound healing [37,38]. Furthermore, when HSCs are treated with a combination of FXa and thrombin, there is an increase in α-SMA, procollagen, TGFβ-1, and significantly improved cell contraction compared with FXa or thrombin alone [39].

Preclinical studies have confirmed the importance of PARs in fibrogenesis, showing how inhibition or lack of PAR 1 and PAR 2 reduces the evolution of liver fibrosis [38,40].

However, this evidence about coagulation factor-mediated HSCs activation by PAR has been based mainly on in vitro research. Poole et al. recently studied the effect on the liver of chronic exposure to carbon tetrachloride (CCl4) in vivo, using a mouse model with PAR 1 deletion specific for HSCs. PAR 1 deletion was linked to decreased activation of HSCs and collagen deposition but was not protective against acute liver damage after CCl4 exposure [41]. Thus, PAR 1 appears to play a role in the “healing process” that occurs after liver injury rather than in its acute phase. Indeed, enoxaparin treatment significantly reduced portal hypertension, hepatic fibrosis, HSCs activation, and desmin expression in mice with CCl4-induced cirrhosis without having any effect on the acute injury. In addition, molecular analysis revealed decreased hepatic fibrin deposition in enoxaparin-treated rats, implying the role of intrahepatic microthrombosis as a primary mechanism of PAR activation [23].

Regarding the etiology of liver fibrosis, thrombin-mediated HSC activation appears to be closely related to the progression of nonalcoholic fatty liver disease (NAFLD) [42]. In addition, administration of a direct thrombin inhibitor to mice with NAFLD reduces HSCs activation, α-SMA expression, and hepatic collagen type 1 mRNA levels [42]. While thrombin has been shown to play a crucial role in the progression of NAFLD, a preclinical investigation suggested that PAR 1, its receptor, is essential for the development of hepatic steatosis in mice fed a Western diet [43]. Nault et al. studied the involvement of PAR 1 signaling in liver damage caused by the contaminant 2,3,7,8-tetrachlorodibenzo-p-dioxin (TCDD). C57BL/6 mice exposed to TCDD acquire NAFLD-like features, such as steatosis, liver damage, inflammation, and fibrosis. Subchronic exposure to TCDD also causes increased intrahepatic coagulation, as reflected by increased thrombin production and deposition of fibrin and fibrinogen in the liver. As measured by serum alanine aminotransferase activity, PAR 1 deficiency had no effect on TCDD-induced hepatocellular damage and hepatic lipid accumulation but nevertheless was linked to a significant reduction in liver fibrosis and histologic evidence of inflammation [44]. 

Although the function of microvascular thrombosis in the evolution of a chronic disease such as NAFLD seems well established, in contrast, the importance of PARs in the progression of chronic viral hepatitis to fibrosis has been less studied. One study examined the PAR 1 genotype in people with chronic HCV infection and found that a specific PAR 1 polymorphism (1426 C/T) was linked to histological evidence of increased liver fibrosis [45]. However, the role of PAR 1 in the progression of viral hepatitis-related liver fibrosis is mainly unknown. The PAR signaling pathway of HSCs is currently believed to be the main element explaining the direct association between microvascular thrombosis and fibrogenesis in the liver parenchyma, although there may be variations depending on the etiology of liver disease. It is also possible to hypothesize a different role of PAR 1 expressed by different cell types; in fact, a number of non-parenchymal liver cells and other cells that infiltrate the damaged liver, including sinusoidal endothelial cells, inflammatory cells (monocytes, neutrophils, and lymphocytes), resident hepatic macrophages (Kupffer cells), and bile duct epithelial cells may express PAR 1 [46,47,48]. Therefore, since microvascular thrombosis may be a manifestation of a harmful hepatic stimulation, PAR 1 activation of HSCs may be the primary cause of the abnormal response leading to fibrosis and its progression. However, considering the wide variety of interactions that occur during hepatic fibrogenesis, it may not be the only one and may indeed be part of a more complex network of molecular and cellular pathways yet to be defined.

## 4. The Role of Liver Sinusoidal Endothelial Cells and Neutrophil Extracellular Traps

Another important cell type involved in the relationship between microvascular thrombosis and fibrosis progression is hepatic sinusoidal endothelial cells (LSECs). It is commonly recognized that LSECs interact with other cells, such as neutrophils, lymphocytes, HSCs, hepatocytes, and Kupffer cells, which is critical in the progression of nonalcoholic steatohepatitis (NASH) to fibrosis [49]. Healthy endothelial cells, on the other hand, express molecules that inhibit platelet activation, coagulation, and thrombosis [50]. Expression of pro- and anti-thrombotic elements changes when LSECs lose their antithrombotic phenotype due to endothelial dysfunction [51,52]. The expression of thrombomodulin, nitric oxide, or prostaglandin I2 is attenuated in the presence of dysfunctional LSECs, which also expose Von Willebrand factor (VWF), integrins, and other receptors that interact with activated platelets and cause clot formation [53,54]. It has also been shown that hepatitis viruses, including hepatitis B virus (HBV) and murine acute hepatitis virus 3, a member of the *Coronaviridae*, can induce LSECs to overexpress Fgl2/fibroleukin prothrombinase, which is crucial for the initiation and progression of fibrin deposition [55,56,57]. Recent histological and molecular studies in a mouse model of congestive hepatopathy demonstrated that LSECs exposed to mechanical stretch upregulate Notch-dependent transcription factors through an integrin-dependent pathway and interaction with the mechanosensitive piezo-calcium channel. As a result, LSECs increase the production of neutrophil chemoattractant C-X-C motif ligand (CXCL) 1, attracting platelets and neutrophils into hepatic sinusoids and leading to the formation of extracellular neutrophil traps (NETs). NETs are complexes consisting of a backbone of extracellular DNA fibers bound to histones and granular proteins, such as myeloperoxidase and neutrophil elastase, which are strongly linked to sinusoidal thrombosis [58]. Interestingly, the progression of NASH has been linked to parenchymal neutrophil infiltration and NET development in mice [59]. In addition, in biopsy samples of the lungs, liver, and kidneys from patients with severe COVID-19, NETs have been linked to microvascular thrombosis [60,61]. As previously shown, when LSECs are dysfunctional, they lose their antithrombotic activity, promoting local thrombotic events and, consequently, the progression of PAR 1-induced fibrosis. In addition, LSECs themselves overexpress PAR 1, which is activated by the coagulation cascade in response to injury, as was demonstrated in a preclinical model of ischemia in mouse liver. In LSECs, PAR 1 activation blocks the ERK1/2 pathway and promotes apoptotic signaling, exacerbating liver damage and causing inflammation and fibrosis [62]. Kruppel-like factor 2 (KFL2), a transcription factor, has recently been recognized as a crucial regulator of endothelium homeostasis in response to inflammatory stimuli, coagulation factors, and hemodynamic stresses such as laminar shear stress [63,64,65]. In LSECs of cirrhotic mice, transcriptome analysis revealed downregulation of Kruppel-like factor (KLF) 2 and 4. In addition, Marrone et al. showed that overexpression of KLF2 in LSECs and HSCs derived from cirrhotic rats reduces HSC activation and enhances paracrine cross-talk between LSECs [66,67]. This is consistent with the decrease in fibrosis and portal pressure associated with KFL2 overexpression in animal studies [68]. Interestingly, KLF2 is primarily activated by the extracellular signal-regulated kinase 5 (MEK5)-extracellular signal-regulated kinase 5 (ERK5) pathway [69], the overexpression of which has recently been linked to suppression of PAR 1 signaling in cell types such as pneumocytes and alveolar barrier endothelial cells [70]. Further research revealed the function of the long noncoding RNA Airn in controlling KLF2. In fact, Airn interacts with subunit 2 of the Polycomb Enhancer Of Zeste 2 repressive complex to maintain the differentiation of LSECs through the KLF2 pathway, preventing the capillarization of LSECs, maintaining the quiescence of HSCs and attenuating the progression of fibrosis. Airn is highly expressed in the liver and serum of patients with fibrosis and in mouse fibrotic livers [71].

## 5. Platelets: Not Just Hemostatic Functions

Endothelial cells, neutrophils, macrophages, HSCs, and clotting factors interact with platelets, which are an essential part of hemostasis [72,73], involved in the development of microvascular thrombi, including hepatic sinusoids [74,75,76]. During the activation phase, platelets can release mediators such as sphingosine-1-phosphate (S-1-P), which activate rat HSCs in vitro but also different subtypes of PAR, including PAR 1 [77,78]. It has been shown that both patients with chronic liver disease and mouse models of liver fibrosis accumulate platelets and the platelet-derived chemokine CXCL4 near fibrotic regions [79]. In vitro, platelet-derived CXCL4 was able to stimulate HSCs proliferation and chemotaxis, while its genetic deletion in mice significantly reduced liver damage and fibrosis [79]. It has also been shown in two mouse models of biliary fibrosis that PDGF-B activates HSCs causing liver fibrosis [80]. Notably, PDFG-B is one of the most effective mitogens for HSCs [81]. In addition, platelets contain high levels of TGF-β1, which is closely associated with the hepatic fibrogenic process [82].

In a mouse model of NAFLD, the expression of neurobeachin-like 2 (NBEAL2) by platelets might play a role in the progression of NAFLD [83]. The NBEAL2 gene is essential for preserving platelet α-granule integrity and is also critical for hemostasis and inflammation [84]. Lower hepatic T-cell and neutrophil infiltration, reduced level of intrahepatic macrophage activation and a slower rate of fibrosis progression were observed in NBEAL2 knockout mouse models of NAFLD [83]. In addition, the platelet adhesion receptor GPIb might be involved in the late fibrogenic phase of NAFLD by interacting with Kupffer cells [83]. Antiplatelet therapy has been associated with a reduction in NAFLD-related fibrosis in several preclinical models, with encouraging results in humans as well [83,85,86,87]. On the other hand, administration of thrombopoietin in a rat model of cirrhosis caused by dimethylnitrosamine provided positive results, reducing the progression of liver fibrosis, while antiplatelet serum reduced this favorable effect [88]. Beyond the specific setting of NAFLD, the role of platelets in the progression of liver fibrosis remains uncertain. In fact, HSC activation can be limited or suppressed through the cAMP pathway, triggered by direct contact with the ATP-enriched granules of adhesive platelets [89]. It has also been reported that platelets could exert an antifibrotic effect by inhibiting HSCs through the release of hepatocyte growth factor (HGF) and activation of Met signaling pathways, resulting in reduced expression of type I collagen genes [90]. Because platelets interact with cellular components, participating in hemostasis, inflammation, fibrogenesis, and tissue regeneration and healing, their function may be ambiguous and dependent on more complex interactions [91]. For example, PAR 4-mediated platelet activation might have a protective effect on the progression of liver fibrosis, as was shown in a mouse model of cholestatic liver injury [92]. In this study, PAR 4-deficient platelet-deficient mice developed more pronounced liver damage, inflammation, and fibrosis following bile duct ligation than wild-type mice. In addition, wild-type mice treated with a PAR 4 antagonist showed a similar increase in liver injury and fibrosis, while treatment with PAR 1 antagonists resulted in opposite effects [92]. However, unlike humans, mouse platelets lack PAR 1, limiting the possibility of defining the role of PAR 1-dependent pathways in liver disease [93].

## 6. ADAMTS 13—Von Willebrand Factor, a Bridge between Coagulation and Fibrosis

Liver cirrhosis has long been considered an acquired bleeding condition because of altered coagulation parameters. Indeed, there is a reduction in liver-related coagulation factors, such as Factors II, V, VII, IX, and XI, but levels of vitamin K-dependent anticoagulant proteins, protein C (PC), protein S, and antithrombin III are also reduced [94,95]. However, it is now recognized that the risk of bleeding is not a consequence of coagulation imbalance but is related to portal hypertension, and that cirrhotic patients maintain a perfect hemostatic balance in most cases [96]. In fact, studies by Tripodi et al. proved that the thrombin generation assay remains preserved in these patients despite these alterations [97,98]. Increased levels of Factor VIII (FVIII) and VWF have also been observed in cirrhotic patients with portal hypertension, suggesting that a procoagulant milieu may prevail [99,100].

These factors are produced and stored in the endothelial cells of portal and hepatic veins but not in hepatic sinusoids [101]. However, in advanced stages of liver fibrosis, endothelial cells of hepatic sinusoids acquire the phenotype of vascular endothelium due to chronic inflammation and endotoxemia, which are the main contributors to portal hypertension [102,103,104,105,106], and begin to produce FVIII and VWF. VWF is a multimeric glycoprotein synthetized and released by vascular endothelial cells in the bloodstream in the form of multimers, with the ability to bind platelets proportionally to their size, demonstrating a pivotal role in hemostatic balance [107]. It is stored in Weibel–Palade bodies (WPBs) and excreted upon stimulation. Some studies have shown a direct correlation between circulating FVIII levels and the severity of portal hypertension as measured by hepatic venous pressure gradient (HVPG) or the presence of ascites and the risk of variceal hemorrhage in patients with advanced liver disease [102,103,104,105,106]. In addition, the association between FVIII alterations and liver fibrosis is well-known: in particular, FVIII and VWF have been observed in capillaries and pericellular regions alongside necrotic sites in the liver parenchyma, and inflammatory injury may promote the deposition of these factors along with fibrosis [108]. A Disintegrin and Metalloproteinase with a Thrombospondin Type 1 motif, member 13 (ADAMTS-13), is another factor influenced by chronic inflammation and advanced liver disease. It is a metalloproteinase that cuts the multimeric VWF between Tyr1605 and Met1606 in its A2 domain. ADAMTS-13 is mainly produced in the liver by HSCs [109,110,111,112,113], and finely regulates hemostatic balance by controlling the size of VWF multimeters, thus their ability to aggregate platelets by forming microthrombi. In chronic liver disease, HSCs that acquire a myofibroblastic phenotype lose the ability to produce and store ADAMTS-13 [111,114,115,116].

Moreover, ADAMTS-13 is inversely correlated with the severity of liver dysfunction in terms of antigen production and activity, whereas VWF antigen and activity increase, as previously discussed, resulting in the release of high molecular weight VWF (HMWVWF) multimers [117].

Therefore, ADAMTS-13 and VWF imbalance may have a pivotal role in the paradigm of parenchymal extinction and liver fibrosis progression in chronic liver diseases [118,119]; indeed, the upregulation of VWF levels in the presence of chronic inflammation, vascular damage, or endotoxemia [102,103] is not counterbalanced due to the deficiency of ADAMTS-13, promoting the formation of platelet microthrombi and fibrin deposition in hepatic sinusoids, with the loss of liver parenchyma and fibrogenesis [120,121].

This condition is similar to thrombotic thrombocytopenic purpura, a primary or acquired clinical disorder induced by the loss of ADAMTS-13 activity or its deficiency that lead to microthrombi formation in small vessels such as glomeruli, cerebral vessels, and cutaneous capillaries [115,121,122,123].

The mechanisms leading to microvascular thrombosis in cirrhosis and the consequent alterations in liver parenchyma are reported in Figure 1.

## 7. Hemostatic Balance in Liver Disease: Translational Implications

### 7.1. Correlation with Liver-Related Outcomes

Since the ADAMTS-13 to VWF ratio (ADAMTS-13/VWF) and FVIII to PC ratio (FVIII/PC) are linked to alterations that occur during liver decompensation or are associated with complications related to portal hypertension, their role as predictive biomarkers has been postulated in several studies.

Kalambosis et al. analyzed FVIII and PC serum levels as markers of coagulation balance in 102 patients with liver cirrhosis and thrombocytopenia. Patients were stratified according to Child–Pugh and Model for End-Stage Disease (MELD) scores. All events related to liver decompensation, including portal vein thrombosis (PVT) or liver transplantation or death, were recorded; during follow-up, 13.7% of patients developed PVT, and those with reduced liver function showed lower levels of FII, V, VII, AT, and PC, but higher levels of FVIII and VWF. FVIII/PC in this group was higher and, together with VWF, was found to be significantly associated with liver disease severity on multivariate analysis. In addition, levels of VWF, FVIII, and FVIII/PC were independently associated with the development of ascites and variceal bleeding. Specifically, VWF levels above 213% were predictive of new-onset ascites, whereas levels above 466% or FVIII/PC above 3.29% predicted variceal bleeding during follow-up. These parameters were also associated with the risk of death from liver-related causes (VWF levels cut-off 392% and FVIII/PC greater than 2.92%) [124].

Schneiner et al. reported that FVIII/PC levels correlated with liver disease severity according to Child–Pugh and MELD scores, and with the presence of portal hypertension, as assessed by the measurement of HVPG. FVIII/PC was significantly higher in the presence of varices, ascites, or hepatic encephalopathy. Markers of inflammation, such as cytokines, and those of liver fibrosis, were also associated with a higher FVIII/PC. FVIII/PC was not associated with increased incidence of bleeding or thrombotic risk on multivariate analysis, but showed a significant association with Child–Pugh and MELD scores, the presence of esophageal varices, and HVPG, demonstrating its strength as a prognostic factor. Schneiner et al. also demonstrated a predictive role of the FVIII/PC ratio in the development of acute on chronic liver failure (ACLF) in patients with decompensated advanced chronic liver disease (ACLD); moreover, FVIII/PC provided prognostic information independently of the chronic liver failure (CLIF) score: patients were stratified according to a cut-off for ACLF development (FVIII/PC > 4.46): patients who presented a baseline level of FVIII/PC > 4.46 presented a higher risk of ACLF development during follow-up [125]. ADAMTS-13/VWF may be another predictor of decompensation or death in cirrhotic patients. In a recent study including 86 patients diagnosed with liver cirrhosis, the lowest levels of ADAMTS-13/VWF and the highest levels of FVIII/PC were found in patients developing decompensation during follow-up. FVIII/PC > 2.6 and ADAMTS-13/VWF < 0.26 were correlated with the risk of hepatic decompensation or death. During multivariate analysis, both indices were independently predictive of disease severity and linked with portal hypertension, with an accuracy comparable to Child–Pugh and MELD scores [126]. In patients with severe alcoholic hepatitis with or without cirrhosis, a negative association between ADAMTS-13, prothrombin time, and total serum bilirubin was observed, while VWF serum level was directly associated with disease severity [119]. ADAMTS-13 was also reduced in patients with decompensated liver cirrhosis and severe alcoholic hepatitis. Non-survivor patients with multi-organ failure showed the highest VWF/ADAMTS-13, while surviving patients improved ADAMTS-13 serum levels during follow-up, with a consensual reduction in HMWVWF. Takaya et al. also reported in a cohort of 99 patients with chronic liver disease, equally distributed according to the Child–Pugh score, that ADAMTS-13 activity and antigen were lower in patients with more severe liver disease, and their alteration was directly related to changes in albumin, prothrombin time, and platelet count [127]. Other studies confirmed that HMWVWF levels were proportionally related to the severity of Child–Pugh and MELD scores and associated with the severity of portal hypertension [128]. VWF also predicted transplant-free mortality in association with C-reactive protein, with significant differences in survival between groups: the 5-year mortality was 34%, 48%, and 72% in the low, intermediate, and high-risk group, respectively, while the probability of liver transplantation increased dramatically at 5 years: 7%, 18%, and 20%. Nevertheless, VWF ristocetinic cofactor (VWF:RCo) > 390% (67% sensitivity and 88% specificity) and ADAMTS-13 activity ≤ 54% (71% sensitivity and 76% specificity) were able to predict poor transplant-free survival [129]. The interaction between hepatic decompensation, coagulation factors alteration, and endotoxemia has also been studied in acute liver failure (ALF). These patients show high endotoxin levels and HMWVWF multimers, as well as low plasma levels of ADAMTS-13 on admission [130]. In a retrospective study including 101 patients divided into a group of 34 subjects with ACLF (cohort named “post ACLF”) and a group of 67 patients without ACLF, 13 of whom developed ACLF during follow-up (cohort named “pre-ACLF”), levels of ADAMTS-13 activity and VWF antigen were assessed at baseline and at the onset of ACLF. In the post-ACLF group, 21 patients died; ADAMTS-13 plasma levels at baseline were higher in survivors than in non-survivors and decreased progressively from patients without ACLF to those with ACLF to patients who resolved ACLF, while VWF antigen showed an opposite trend. During multivariate analysis, ADAMTS-13/VWF remained an independent prognostic factor for ACLF onset in patients with a previous episode of acute on chronic liver decompensation [131].

In another study, VWF and ADAMTS-13 serum levels were assessed at admission and daily for one week in a cohort of 676 patients with drug-induced acute liver injury. Among them, 483 survived without liver transplantation 21 days after enrollment; patients who developed ALF had three-fold higher HMWVWF multimers than controls, while ADAMTS-13 activity was four-fold lower. Patients with severe hepatic encephalopathy, systemic inflammatory response syndrome, or acute kidney injury had the lowest levels of ADAMTS-13 combined with the highest levels of VWF. No difference in thrombin generation was documented. Interestingly, in patients with bleeding complications, ADAMTS-13/VWF was reduced compared with other patients. Furthermore, low ADAMTS-13 activity was associated with more severe hepatic encephalopathy, a high risk of liver transplantation, and death from hepatic causes [132].

Taken together, these data confirm that changes in coagulation parameters reflect the risk of liver disease progression and decompensation without any influence on the hemostatic balance. The main studies showing the role of these molecules as potential biomarkers of liver disease progression are reported in Table 1.

### 7.2. The Gut–Liver Axis Is Associated with Alteration of Hemostatic Factors

Because liver cirrhosis evolution is strongly influenced by gut dysbiosis and bacterial translocation, low-grade chronic inflammation, and endotoxemia, several authors evaluated the association between markers of intestinal health and coagulation parameters [133]. Takaya et al. found an association between endotoxin levels and the presence of HMWVWF multimers [130]; in addition, VWF and FVIII/PC correlated with lipopolysaccharide-binding protein (LBP) and IL 6 plasma concentration. A multivariate analysis excluded the influence of portal hypertension on these alterations, clarifying the direct effect of endotoxemia in this process. In addition, VWF serum levels independently predicted variceal hemorrhage, paracentesis requirement, and risk of spontaneous bacterial peritonitis and other bacterial infections, but did not correlate with the occurrence of hepatic encephalopathy [128].

Carnevale et al. confirmed the relationship between circulating lipopolysaccharides (LPS) and alterations in FVIII and VWF in patients with chronic liver disease. LPS serum levels also correlated with *Escherichia coli* circulating DNA, suggesting a modulating effect of microbiota-derived LPS on endothelial cells releasing VWF and FVIII. The same group developed human endothelial cell cultures to be exposed to a range of increasing concentrations of LPS derived from the serum of cirrhotic patients, ultimately obtaining FVIII and VWF secretion from WPBs [134]. Thus, this study confirmed that low-grade endotoxemia induced by the translocation of bacteria from the intestinal lumen to the portal and systemic circulation is able to alter the balance of coagulation factors. Our group further evaluated changes in ADAMTS-13/VWF in patients with advanced chronic liver disease in relation to markers of endotoxemia such as LPS, confirming the strong association between decreased ADAMTS-13/VWF, chronic low-grade inflammation, and risk of liver failure. In fact, ADAMTS-13/VWF was significantly lower in patients who developed complications (ascites, variceal hemorrhage, and hepatocellular carcinoma) than in those who did not, while LPS showed an opposite trend. In addition, the analysis of the gut microbiota composition of these patients showed a depletion of *Akkermansia* in stool samples, associated with a specific metabolic profile; the loss of *Akkermansia* may increase intestinal permeability exacerbating gut-derived inflammation, with the consequent modulation of ADAMTS-13/VWF and development of chronic liver disease complications [135]. Other studies reported an inverse correlation between ADAMTS-13 activity and endotoxemia, as opposed to VWF, which shows a direct correlation, resulting in an imbalance between ADAMTS-13 and VWF, as previously described [132]. Ishikawa et al. also observed that the increase in VWF/ADAMTS-13 in patients with severe alcoholic hepatitis was associated with increased production of inflammatory cytokines, such as TNF-α, IL6, IL8, and plasma endotoxin, the latter being inversely correlated with ADAMTS-13 activity [136]. Finally, in a previously mentioned study by Reuken et al., patients with overt systemic inflammation and bacterial translocation had higher levels of VWF, which correlated with leukocytes blood count, C Reactive Protein (CRP) and LBP serum concentrations, as well as acute kidney injury during follow-up, also being predictive of liver-related death and 2-year transplant-free survival [129].

## 8. Therapeutic Perspectives

The strict association between coagulation imbalance and liver disease progression pushed us to analyze the impact of anticoagulants and antiplatelet agents on fibrogenesis. Human studies in this field are scant, and data from animal models are encouraging although preliminary; the main data are reported in Table 2 and discussed in the following paragraphs.

### 8.1. Low Molecular Weight Heparin

Because parenchymal extinction induced by microthrombosis is a major factor in liver fibrogenesis, preclinical studies aimed to evaluate whether anticoagulants and antiplatelet drugs can prevent the progression of liver damage. In a preclinical model of rats with CCl4-induced liver damage, administration of low molecular weight heparin (LWMH) induced a significant improvement in the severity of fibrosis, and after hepatectomy, the treated group showed a reduction in total serum bilirubin [137]. In addition, enhanced liver regeneration was observed in a group of CCl4-treated rats after treatment with deltaparin, the effects of which were attributed to the stimulation of HGF and inhibition of HSCs [138]. In a mice model of damage caused by bile duct ligation, enoxaparin administration led to an improvement in cytonecrosis indices, and, on histological analysis, liver necrosis and fibrosis were significantly reduced in treated rats compared with controls [139]. Another study aimed to evaluate the long-term effects of enoxaparin administration on liver fibrosis and hemodynamic changes in two rat models of cirrhosis (CCl4-induced and thioacetamide [TAA]-induced). After one week of enoxaparin injection, portal pressure was reduced in association with a decrease in hepatic vascular resistance, superoxide, and nitrotyrosine concentration; this was coupled with a 26% reduction in fibrosis area, in line with decreased expression of α-SMA, platelet-derived growth factor receptor beta (PDGFR beta), and procollagen I by HSCs. Long-term administration of enoxaparin was associated with a marked reduction in hepatic venule resistance, portal pressure, and liver tissue fibrosis than in untreated controls, with a reduction in microthrombi formation. Furthermore, enoxaparin administration did not cause alterations in liver enzymes in these rats [23]. A metanalysis analyzed 16 studies reporting on the role of anticoagulants used in animal models of chronic liver diseases, confirming a significant improvement in fibrosis deposition according to changes in METAVIR fibrosis score in cytonecrosis and inflammatory parameters and liver function [140]. The only study on the administration of prophylactic enoxaparin in patients with cirrhosis was performed in a cohort of Child–Pugh B-C patients. It was a double-blinded prospective case–control trial including only patients without episodes of hepatic decompensation in the previous 3 months, without evidence of PVT or splenomesenteric thrombosis, and without increased bleeding or thrombotic risk factors. Enoxaparin 4000 UI/die was administered in the treatment group for 48 weeks; the primary endpoint was 2-year prevention of PVT or mesenteric vein thrombosis, whereas the occurrence of liver decompensation, overall survival, and transplant-free survival were the secondary endpoints. Seventy patients were enrolled, and thirty-four received enoxaparin. After 2 years of follow-up, no patients in the enoxaparin group developed PVT, compared to 27.7% in the control group. Treatment was associated with lower rates of hepatic decompensation with no increased risk of variceal bleeding. Liver function tests were significantly improved in the enoxaparin group at 48 weeks of follow-up, and higher survival was reported compared to controls. Interestingly a reduction in inflammatory markers related to endotoxemia was described. The authors hypothesized a beneficial effect of enoxaparin in preventing intestinal microthrombosis and inflammation, drivers of bacterial translocation in liver cirrhosis [141,142,143]. A few studies did not confirm the beneficial effects of LMWH on liver fibrogenesis. For instance, Fortea et al. [144] demonstrated that enoxaparin did not improve hepatic outcomes in three rat models of cirrhosis induced by chemicals and cholestasis. The administration of enoxaparin at prophylactic and anticoagulant doses after an acute liver injury did not show any beneficial effect on fibrosis and cirrhosis progression. One important aspect to consider is the timing of enoxaparin administration in relation to the stage of cirrhosis. In most cases, enoxaparin was administered when cirrhosis was already in an advanced stage. This could have influenced the outcomes, as the effectiveness of enoxaparin might differ depending on the stage of the disease. It is possible that earlier administration before the acute injury occurred, could have yielded different results. Nonetheless, the role of enoxaparin in the context of microvascular thrombosis is more ambiguous in rats, as their platelets lack PAR 1. These results could suggest the existence of a therapeutic window in which anticoagulants may prevent liver fibrosis progression.

### 8.2. Direct Oral Anticoagulants

There are very few studies on the use of direct oral anticoagulants (DOACs) in liver cirrhosis. In preclinical models of TAA-injured rats, administration of dabigatran, an anti-Xa factor, reduced collagen and fibrin deposition in the liver [145]. Another study reported that rats treated with CCl4 and rivaroxaban, another anti-Xa agent, showed significant preservation of biochemical parameters, including inflammatory and fibrosis markers [146]. Male rats were randomized into three groups: CCl4 fibrotic rats that received or did not receive rivaroxaban 5 mg/kg, and a control group; rivaroxaban treatment reduced fibrosis markers, tissue factor, fibrin, and α-SMA levels in liver tissue, suggesting a role in attenuating CCL4-induced liver damage.

Similar results were obtained in a recent study on CCl4 and TAA rat models of cirrhosis [147]. Animals treated with rivaroxaban 20 mg/kg/day achieved a significant reduction in portal pressure and vascular resistances but no change in systemic hemodynamics. Reduction in portal pressure and hepatic vascular resistances as well as of the expression of profibrotic factors α-SMA, collagen type I alpha 1 chain (COL1A1), PDGF beta, tissue inhibitor metallopeptidase 1/2 (TIMP1/2), and TGFβ were observed, with improved intraparenchymal fibrin and collagen deposition. These results confirm the close interaction between microthrombosis and hepatic fibrosis, also supported by the reduction of VWF in the CCl4-treated group. In contrast to enoxaparin, cirrhotic rats receiving rivaroxaban show a greater response to acetylcholine stimulation, suggesting improved endothelial function and, in addition, rivaroxaban induced an anti-inflammatory phenotype in HSCs, reducing their activation [24].

### 8.3. Antiplatelet Agents

The antifibrotic properties of aspirin can be attributed to its antithrombotic activity, which may counterbalance platelet hyperaggregability induced by chronic low-grade inflammation. In addition, aspirin has anti-inflammatory properties that stabilize endothelial cells integrity, reduce the release of inflammatory chemokines and interleukins, and via nuclear factor kappa-light-chain-enhancer of activated B cells (NF-kB), inhibits the production of TGF-beta, which is a promoter of fibrogenesis [148]. In a preclinical model of TAA-induced cirrhosis, aspirin-treated rats showed a reduction in mortality compared to controls, associated with a significant improvement in the degree of fibrosis [137]. After hepatectomy, liver regeneration was enhanced and fibrosis reduced, although the mechanism is unclear. In another animal model, administration of low or high doses of aspirin partially prevented acute and chronic TAA-induced liver damage, with a reduction of transaminases and total bilirubin serum levels. After 8 weeks, significant differences in terms of hepatic fibrin and collagen deposition were recorded in the control group compared to the aspirin-treated group (*p* < 0.05) [24]. Sitia et al. showed that mice infected with HBV and treated with antiplatelet drugs (aspirin or clopidogrel) had mild signs of chronic hepatitis with poor collagen deposition; none of the mice developed cirrhosis, and a significantly lower percentage of advanced fibrosis was found in the antiplatelet-treated group [85]. At liver histology, treated mice had no platelet aggregates in the vessels outside the necroinflammatory areas. Similar results were obtained in a preclinical model of NAFLD treated with aspirin, ticlopidine, or cilostazol, with a reduction in inflammatory cells and inhibition of procollagen proteins compared with controls, with improvement in hepatic steatosis, inflammation, and fibrosis [149]. Comparing anti-factor Xa and anti-PYP12 clopidogrel, the antiplatelet drug was more effective in preventing inflammation and fibrosis in a preclinical model of CCL4 liver injury, with a marked reduction in TGFbeta and α-SMA [147]. No studies in human subjects have directly evaluated the effect of antiplatelet drugs on fibrosis, but indirect data can provide some evidence. In a retrospective study including 180 patients with recurrent chronic HCV infection after liver transplantation, aspirin intake emerged as an independent protective factor against the development of fibrosis 1 year after surgery [150]. In a larger prospective study of patients with chronic hepatitis C or B, including 50,275 adults, 14,205 patients were taking aspirin; a lower incidence of hepatocellular carcinoma in association with lower liver-related mortality was demonstrated in this subgroup [151].

### 8.4. Statins

Statins have pleiotropic effects and affect liver fibrosis in several ways. In human cohorts of cirrhotic patients, retrospective studies show that patients taking statins have a lower likelihood of hepatic decompensation during follow-up [152]. A study including human subjects with HCV-related cirrhosis reported that liver fibrosis progressed during follow-up in only 10% of patients treated with statins compared with 29% of controls, and the association remained significant after correction for other parameters [152,153,154]. Previous investigations reported that statins could reduce portal hypertension, as assessed by HVPG, in cirrhotic patients, although specific histologic data are lacking [155]. What is known about the effects of statins administration on hepatic function has been obtained from preclinical models; a decrease in the expression of NF-kB resulting in reduced reactive oxygen species and proinflammatory cytokines release was observed, with a protective effect on the endothelium by reinforcing the integrity of hepatic sinusoids [156,157].

Simvastatin also improved portal hypertension in rat models of ACLF, decreasing LPS-induced HSCs activation and improving survival [158]. Nevertheless, simvastatin prevents liver damage and microthrombi formation induced by the administration of LPS 5 mg/kg in animals; simvastatin also reduces intrasinusoidal fibrin deposition and preserves sinusoidal thrombomodulin expression [159]. Table 2 briefly summarizes the main knowledge on anticoagulants, antiplatelet drugs, and liver fibrogenesis.

**Table 2 cells-12-01712-t002:** Studies reporting the effect of anticoagulants, antiplatelet agents and statins on liver fibrosis progression.

Study	Model	Drug	Results
Assy et al., 2007 [137]	TAA-induced liver damage in rats	Enoxaparin	↓ liver fibrosis severity (METAVIR score), ↓ total serum bilirubin, ↑ liver regeneration
Abe et al., 2006 [138]	CCl4-induced liver damage in rats	Dalteparin	↓ liver fibrosis progression, ↑ hepatocyte growth factor (HGF), inhibition of HSCs, ↑ liver regeneration
Abdel-Salam et al., 2005 [139]	Bile duct ligation in mice	Enoxaparin	↓ liver necrosis
			↓ fibrosis
Cerini et al., 2016 [23]	CCl4- and TAA-induced cirrhosis in rats	Enoxaparin	↓ hepatic venule resistance, ↓ portal pressure, ↓ hepatic fibrin deposition, ↓ HSCs activation, ↓ liver fibrosis
Villa et al., 2012 [141]	Patients with cirrhosis (Child–Pugh B-C)	Enoxaparin (4000 UI qd) for 48 weeks	No patients in the treatment group developed PVT, ↓ decompensation, ↑ liver function, ↑ survival, ↓ inflammatory markers
Lee et al., 2018 [145]	TAA-induced liver damage in rats	Dabigatran	↓ collagen and fibrin deposition in the liver
Mahmoud et al., 2019 [146]	CCl4-induced liver damage in rats	Rivaroxaban	↓ inflammatory and fibrosis markers, ↓ liver fibrosis
Assy et al., 2007 [137]	TAA-induced liver damage in rats	Aspirin	↓ liver fibrosis severity (METAVIR score), ↓ total serum bilirubin, ↑ liver regeneration
Sitia et al., 2012 [85]	Mice infected with HBV	Antiplatelet drugs (aspirin or clopidogrel)	↓ intrahepatic inflammatory cells, ↓ liver fibrosis severity, ↓ HCC risk
Fujita et al., 2008 [149]	Murine model of NAFLD	Aspirin, ticlopidine, or cilostazol	↓ inflammatory cells, ↓ procollagen proteins, ↓ hepatic steatosis, ↓ inflammation, ↓ fibrosis
Poujol-Robert et al., 2016 [150]	Retrospective study in patients with recurrent HCV infection after liver transplantation	Low-dose Aspirin	↓ fibrosis progression

Table 2 abbreviations (in order of appearance): TAA: Thioacetamide, METAVIR: meta-analysis of histological data in viral hepatitis, CCl4: carbon tetrachloride, HGF: hepatocyte growth factor, HSC: hepatic stellate cells, qd: every day, PVT: portal vein thrombosis, HBV: hepatitis B virus, HCC: hepatocellular carcinoma, NAFLD: non-alcoholic fatty liver disease. ↓ reduced, ↑ increased.

## 9. Discussion

Liver fibrosis progression depends on complex cellular and molecular pathways. The high morbidity and mortality associated with chronic liver disease make it mandatory to understand the mechanisms underlying fibrogenesis. Hemostasis plays a primary role in liver fibrosis, both as a consequence of inflammatory response and following sinusoidal blood flow alterations. In this process, HSCs, LSECs, and platelets play pivotal roles. Some molecular pathways may explain the complex interconnections among these cell types in response to sinusoidal thrombosis. In particular, PARs signaling appears to be crucial in the cross-talk that leads from microvascular thrombosis to liver fibrosis. NETs could be the initial promoters of fibrogenesis following hepatic sinusoidal mechanical stretch. In addition, the role of coagulation factors alteration and its strict association with endotoxemia and, more in general, with the gut–liver axis derangement underscores not only their potential as prognostic markers for disease severity and complications but also as a promising field to be further investigated for both pre-emptive or therapeutic purposes. Indeed, it is crucial to note that the precise relationship between all the pieces of this puzzle still remains to be fully elucidated, with further research needed to validate their use for risk stratification and prognostic purposes. Expanding histological evidence, including different demographic groups and patients with various etiologies of liver disease, to be correlated with gut microbiota profiles, would provide a more comprehensive understanding of the role of sinusoidal thrombosis in hepatic fibrosis. In addition, exploring the specific mechanisms through which PARs interacting with HSCs, endothelial cells, and immune cells contribute to thrombosis and fibrogenesis in the liver could shed light on their intricate signaling pathways in response to sinusoidal thrombosis. Additionally, another important goal will be to identify new markers associated with microvascular thrombosis that could predict liver fibrosis progression. The promising results from preclinical and clinical studies on the use of anticoagulants, antiplatelet agents, and other pharmacological interventions to halt liver injury and fibrosis development highlight an unexpected potential. Yet, it is essential to validate these findings in large-scale, well-designed clinical trials to ensure their safety and efficacy in human subjects with advanced chronic liver disease.

## 10. Conclusions

In conclusion, the relationship between coagulation, inflammation, and fibrosis in the context of liver disease requires a comprehensive understanding of the intricate molecular and cellular pathways involved. Indeed, hepatic sinusoidal thrombosis has been associated with the development of liver fibrosis and cirrhosis. At the molecular level, the activation of PARs in HSCs and LSECs may contribute to this process. Additionally, immune cells such as neutrophils and the formation of NETs may also play a role. The involvement of platelets in this complex relationship remains ambiguous and requires further investigation to fully understand their contribution. Coagulation factors, including FVIII and other molecules involved in hemostasis, such as VWF, have been implicated in the progression of liver fibrosis, while factors such as ADAMTS13 have shown potential in reducing fibrogenesis. Gut dysbiosis, which is characterized by a tendency toward procoagulant phenotypes, may have a significant impact on the development and progression of fibrosis. These pieces of evidence, although mainly derived from preclinical models, suggest that anticoagulant and antiplatelet therapies could potentially improve or halt the progression of liver fibrosis. Further research is warranted to unravel the underlying mechanisms and explore potential therapeutic strategies. By deepening knowledge of the molecular pathways of coagulation, inflammation, and fibrosis, novel therapeutic targets and reliable biomarkers can be identified, with the aim to develop effective interventions and improve the management of liver cirrhosis as a chronic disease, ultimately leading to improved patient outcomes.

## Figures and Tables

**Figure 1 cells-12-01712-f001:**
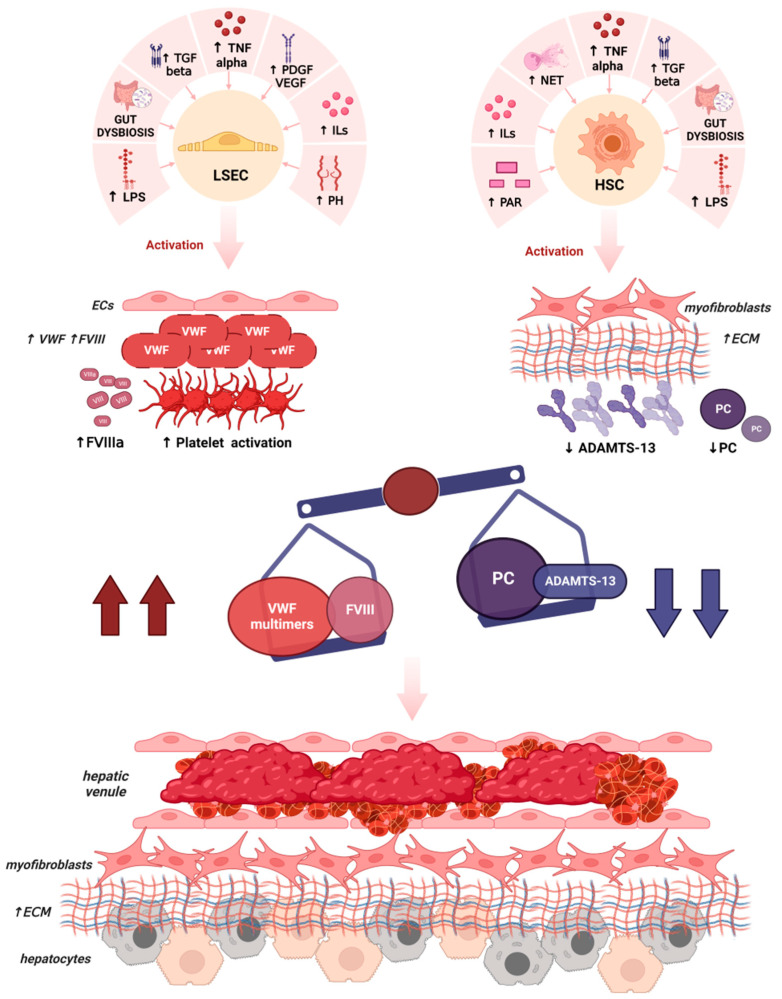
Main mechanisms leading to microvascular thrombosis and parenchymal extinction in liver cirrhosis. In the presence of low-grade chronic inflammation, liver sinusoid endothelial cells acquire vascular endothelial cells phenotype, upregulating the production and release of VWF in the form of HMWVWF multimers and FVIII. Several mediators contribute to this process, such as interleukins (IL6, IL8), chemokines (TNF), growth factors (PDGF, VEGF, and TGF beta), gut dysbiosis, and gut microbiota-derived products (LPS). The same and other factors, such as NET and PAR 1, may also induce significant alterations in HSCs, leading to a myofibroblastic phenotype with expression of alpha-SMA, procollagen, and a poor ability to produce ADAMTS-13, the protein that cleaves the HMWVWF multimers in normal conditions. In liver disease, lower levels of other anticoagulants, such as PC, are also reported. As a result, there is an imbalance between procoagulant factors such as FVIII and VWF and anticoagulant factors such as ADAMTS-13 and PC. HMWVWF multimers, not inhibited by ADAMTS-13, bind activated platelets forming microthrombi. Microthrombi deposition prevalently occurs near the small branches of the hepatic venules, causing the destruction of hepatocytes in a process named “parenchymal extinction”, leading to the worsening of liver fibrosis.

**Table 1 cells-12-01712-t001:** Studies reporting the relationship between ADAMTS-13, VWF, FVIII, and PC and liver-related outcomes.

Study	Design	Clinical Setting	Marker	Outcome
Kalambosis et al. [124]	Observational	Liver cirrhosis with thrombocytopenia	FVIII/PC and VWF Ag	↑ liver-related death (VWF Ag cut-off 321% FVIII/PC cut-off 2.36%) ↑ variceal bleeding (VWF Ag cut-off 466% FVIII/PC cut-off 3.29%) ↑ new-onset ascites (VWF Ag cut-off 213% FVIII/PC cut-off 1.99%) ↑ portal vein thrombosis
Schneiner et al. [125]	Observational	Liver cirrhosis with portal hypertension	FVIII/PC	↑ MELD score ↑ HVPG ↑ ACLF (FVIII/PC cut-off > 4.46)
Ponziani et al. [126]	Observational	Liver cirrhosis	↓ ADAMTS-13/VWF and ↑ FVIII/PC	↑ decompensation rate and risk of liver-related death (FVIII/PC cut-off > 2.6 ADAMTS-13/VWF cut-off < 0.26)
Matsuyama et al. [119]	Observational	Alcoholic hepatitis	↓ ADAMTS-13 activity and ↑ VWF Ag	↑ risk of severe alcoholic hepatitis ↓ survival
Takaya et al. [127]	Observational	ACLF	↓ VWF Ag, ↑ ADAMTS-13 activity	↑ survival (VWF/ADAMTS-13 cut-off < 9)
Driever et al. [132]	Observational	Drug-induced acute liver injury	↑ VWF Ag, ↓ ADAMTS-13 activity	↑ hepatic encephalopathy ↑ bleeding complications ↑ acute kidney injury predictors of the need for liver transplantation

Table 1 abbreviations (in order of appearance): FVIII: factor VIII, PC protein C, vWF Ag: von Willebrand Factor Antigen, MELD Model End Stage Liver Disease; HVPG Hepatic Venous Pressure Gradient, ACLF acute on chronic liver failure, ADAMTS-13, a disintegrin and metalloproteinase with a thrombospondin type 1 motif, member 13. ↓ reduced, ↑ increased.

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
