# Peer review of "Microvascular Thrombosis and Liver Fibrosis Progression: Mechanisms and Clinical Applications"

_cells, 2023, doi:10.3390/cells12131712_

Round 1

Reviewer 1 Report

In this report, the authors review the relationship between microvascular thrombosis and fibrosis progression in liver diseases, with emphasis on cells subpopulations involved in such progression (HSCs, LSECs, neutrophils) and platelets; coagulation factors (ADAMTS-13/VWF), hemostatic factors umbalance and liver diseases (ADAMTS-13/VWF and FVIII/PC ration with cirrhosis, PVT, portal hypertension, ALF, drug-induces liver injury, relationship with gut-liver axis) and therapeutic perspectives (mainly on experimental models and the few published studies in human being).

The topic is interesting and the distribution of the topics is adequate. This report has interest for readers. Nevertheless, I would suggest some minor points to be addressed by the authors prior to consider this draft for publication.

- Point 7.1: in this chapter there is a extensive relation of publications in which ADAMTS-13/VWF and FVIII/PC seem to have clinical relevance in liver disease (mainly focused in cirrhosis , but also include ALF, liver transplant, drug-induced liver injury, hepatic encephalopathy...). The problem with this chapter is the high amount of data presented cause that the reader can get lost. It may be appropriate if authors resume this chapter introducing a table, in a similar way than Table 1 of pages 13/14.

-  Discussion: In opinion of the authors and taking into account of all presented data, what should be the future directions in research to improve the knowledgment of the topic exposed in this review?

- Conclusions: in addition of further studies are needed to elucidate the molecular mechanisms, conclusion should also include the main evidences exposed in the review which may justify such studies.

Minor points:

-Regarding reference 23 (line 108) , there is a study (Fortea et al. Liver International 2018; 38: 102-111) which state that the use of enoxaparin does not ameliorate liver fibrosis in liver cirrhosis in a rat model. What is a possible explanation of such controversy? 

- Line 315: did you mean "Disintegrin"?

- Lines 341-361: seems to be the footnote of Figure 1 but as is stated in the text, appear to be main text. Please, arrange.

- Line 594: arrange the heading.

-

Author Response

" In this report, the authors review the relationship between microvascular thrombosis and fibrosis progression in liver diseases, with emphasis on cells subpopulations involved in such progression (HSCs, LSECs, neutrophils) and platelets; coagulation factors (ADAMTS-13/VWF), hemostatic factors umbalance and liver diseases (ADAMTS-13/VWF and FVIII/PC ration with cirrhosis, PVT, portal hypertension, ALF, drug-induces liver injury, relationship with gut-liver axis) and therapeutic perspectives (mainly on experimental models and the few published studies in human being).

The topic is interesting and the distribution of the topics is adequate. This report has interest for readers. Nevertheless, I would suggest some minor points to be addressed by the authors prior to consider this draft for publication."

We thank the Reviewer for his/her general opinion on the manuscript.

- "Point 7.1: in this chapter there is a extensive relation of publications in which ADAMTS-13/VWF and FVIII/PC seem to have clinical relevance in liver disease (mainly focused in cirrhosis , but also include ALF, liver transplant, drug-induced liver injury, hepatic encephalopathy...). The problem with this chapter is the high amount of data presented cause that the reader can get lost. It may be appropriate if authors resume this chapter introducing a table, in a similar way than Table 1 of pages 13/14."

Thanks for this comment. We reported a simplified table as suggested (Table 1, lines 364-367 of the manuscript).

-  "Discussion: In opinion of the authors and taking into account of all presented data, what should be the future directions in research to improve the knowledgement of the topic exposed in this review?"

Thanks for this comment. As requested, we added a paragraph in the discussion section (lines 536-542)

- "Conclusions: in addition of further studies are needed to elucidate the molecular mechanisms, conclusion should also include the main evidences exposed in the review which may justify such studies".

We include these information in the conclusion section, as suggested (lines 548-562)

Minor points:

-"Regarding reference 23 (line 108) , there is a study (Fortea et al. Liver International 2018; 38: 102-111) which state that the use of enoxaparin does not ameliorate liver fibrosis in liver cirrhosis in a rat model. What is a possible explanation of such controversy?" 

Thanks to the Reviewer for this suggestion. We are glad to propose our point of view: we hypothesized that timing of enoxaparin administration has to be related with the stage of cirrhosis: in most studies, enoxaparin was administered when cirrhosis was already in an advanced stage. This could have influenced the outcomes, as the effectiveness of enoxaparin might differ depending on the stage of the disease. It is possible that earlier administration, before the acute injury occurred, could have yielded different results. Nonetheless, the role of enoxaparin in the context of microvascular thrombosis is more ambiguous in rats, as their platelets lack PAR 1. We included this comment in the manuscript (lines 441-451)

- "Line 315: did you mean "Disintegrin"?"

We apologize for this error. We have revised the text and corrected typos.

- "Lines 341-361: seems to be the footnote of Figure 1 but as is stated in the text, appear to be main text. Please, arrange".

The text was amended accordingly.

 "Line 594: arrange the heading".

It has been corrected in the text.

Reviewer 2 Report

This is a very well-written review of an important topic

It covers the topic in detail. Figures and table are ok

I have no further suggestions

Author Response

"Reviewer nr 2: This is a very well-written review of an important topic. It covers the topic in detail. Figures and table are ok. I have no further suggestions"

We thank the Reviewer for his/her comment. We hope this review will be useful for readers.